DNA metabarcoding reveals diet diversity and niche partitioning by two sympatric herbivores in summer

Li Ruofei 1
Wang Dandan 1
Cao Zhiming 1
Liu Yuqin 1
Wu Wenguo 2
Liu Wuhua 2
Zhan Jianwen 2
http://orcid.org/0009-0002-3159-1133 Xu Yongtao 1 ytxu666@163.com
1 Jiangxi Provincial Key Laboratory of Conservation Biology, Jiangxi Agricultural University , Nanchang, Jiangxi , China
2 Taohongling Sika Deer National Nature Reserve Administration , Pengze, Jiangxi , China
Schuster Richard
Electronic publication date: 2024 Dec 23
Publication date: 2024
Volume: 12
Electronic Location ID: e18665
Received 2024 May 16; Accepted 2024 Nov 18
Copyright: © 2024 Li et al.
Copyright year: 2024
Copyright holder: Li et al.
License: This is an open access article distributed under the terms of the Creative Commons Attribution License, which permits unrestricted use, distribution, reproduction and adaptation in any medium and for any purpose provided that it is properly attributed. For attribution, the original author(s), title, publication source (PeerJ) and either DOI or URL of the article must be cited.
License URL: https://creativecommons.org/licenses/by/4.0/

Keywords: DNA metabarcoding, Herbivores, Dietary partitioning, Niche overlap, Summer

Funding: National Natural Science Foundation of China 32470552 and 31960118 Natural Science Foundation of Jiangxi Province of China 20242BAB25349 This work was supported by the National Natural Science Foundation of China (Project Nos. 32470552 and 31960118) and the Natural Science Foundation of Jiangxi Province of China (No. 20242BAB25349). The funders had no role in study design, data collection and analysis, decision to publish, or preparation of the manuscript.

==============================
Background

Food provides essential nutrients and energy necessary for animals to sustain life activities. Accordingly, dietary niche analysis facilitates the exploration of foraging strategies and interspecific relationships among wildlife. The vegetation succession has reduced understory forage resources (i.e., shrubs and herbs) available to sika deer (Cervus nippon kopschi). Little is known about the summer foraging strategies or the interspecific relationship between sika deer and Reeves’ muntjac (Muntiacus reevesi).

Methods

The present study used high-throughput sequencing and DNA metabarcoding techniques to investigate the feeding habits and interspecific relationships between sika deer and Reeves’ muntjac in our study.

Results

A total of 458 amplicon sequence variants (ASVs) were identified from fecal samples, with 88 ASVs (~19.21%) unique to sika deer and 52 ASVs (~11.35%) unique to Reeves’ muntjac, suggesting the consumption and utilization of specific food items for the two species. The family Rosaceae was the most abundant for both species, especially Rubus spp. and Smilax china. Alpha diversity (local species richness) indicated that the dietary species richness of sika deer was higher than that of Reeves’ muntjac, but the difference was not statistically significant. Sika deer also exhibited a higher evenness index (J′ = 0.514) than Reeves’ muntjac (J′ = 0.442). Linear discriminant effect size analysis revealed significant differences in forage plants between the two herbivores. The niche breadths of sika deer and Reeves’ muntjac were 11.36 and 14.06, respectively, and the dietary niche overlap index was 0.44. Our findings indicate the diet partitioning primarily manifested in the differentiation of food items and the proportion, which ultimately reduces the overlap of nutritional niches and helps avoid conflicts resulting from resource utilization. This study provides a deeper insight into the diversity of foraging strategies and the interspecific relationship of herbivores from the food dimension.

Introduction

Food provides animals with the necessary energy and nutrients for their life activities and as such is a crucial resource for maintaining the survival and growth of populations (Zhang et al., 2020a). Large herbivorous animals are in decline, making them the most endangered group of vertebrates due to habitat fragmentation, climate change, exotic invasive species, artificial disturbance, particularly food resource changes (Kowalczyk et al., 2011; Atwood et al., 2020). In 2020, some wild Asian elephants roamed north from Xishuangbanna National Nature Reserve (China) to search for foraging areas (Jiang et al., 2023). Diet analysis can be an important first step for conserving wildlife by assessing the nutrition intake, exploring the relationship between foraging behavior and habitats, and clarifying the effect of food on intra- and interspecific relationships through qualitative and quantitative analyses (Kartzinel et al., 2015). Such knowledge can further be used to reveal the adaptive mechanisms toward temporal and spatial variation in food availability or diet specialization (Leigh, Papastamatiou & German, 2018; Zhang et al., 2018; Vesterinen et al., 2016).

The sika deer (Cervus nippon) is an endemic ungulate of the East Asian monsoon region. Natural populations of sika deer are distributed over northeastern Asia from the Ussuri region of Russia to mainland China, North Vietnam, Taiwan, and Japan (Tamate et al., 1998). It was classified in 2015 as a Least Concerned species by the International Union for Conservation of Nature (Harris, 2015). In Japan, the number of sika deer declined approximately tenfold from 1990 to 2014, with the current population estimated to be 3.05 million animals (Kawarai et al., 2022). Historically, there were six subspecies of wild sika deer in China that were widely distributed in northeastern, northern, central, southern, and southwestern China and the eastern parts of the Qinghai-Tibet Plateau (Su et al., 2023; Guo & Zheng, 2000). However, by the 1960s only three subspecies remained, the Sichuan sika deer (Cervus nippon sichuanicus), Dybowski’s deer (Cervus nippon hortulorum), and the South China sika deer (Cervus nippon kopschi) (Sheng, 1992). Nowadays, the distribution areas are small and isolated, genetic exchange of populations is limited (Zhang et al., 2016). The population number of sika deer have been decreasing, and the total number of wild sika deer in China is less than 2,000. Thus, this endangered species has been classified as a national Class I protected animal (Blouch et al., 1998; Guo & Zheng, 2000). Taohongling National Nature Reserve (hereafter, TNNR) was established in 2001 to protect the South China sika deer. The sika deer prefers to live in scrub-grassland habitats. Carrying capacity and reduced understory forage have been observed due to vegetation succession. Consequently, sika deer forage beyond the reserve boundary frequently, this poses challenges to wildlife conservation. Management measures to control the successional stage are necessary, i.e., artificial vegetation dwarfing (Jiang, 2009; Jiang et al., 2012; Zou & Liu, 2024), see Fig. 1.

Figure 1 Ecological photographs from the reserve.

(A) Suitable scrub-grassland habitat of sika deer; (B) reduced understory food resources available to sika deer. (C) Vegetation dwarfing experimental plot; (D) comparison between the dwarfed experimental plots and the non-dwarfed areas.

South China sika deer ruts from August to November, usually a singleton pregnancy with a lactation period up to 6 months. Parturition is concentrated between May and July each year when sika deer require considerable energy to raise their offspring (Jiang, 2009). Reeves’ muntjac (Muntiacus reevesi) is a closely related species that coexists with sika deer in the TNNR. Reeves’ muntjac reaches sexual maturity at 7–8 months of age and has a gestation period of 18 weeks. The females can conceive 3–4 days after giving birth, and lactation does not affect their ability to reproduce. Numerous monitoring surveys (i.e., using camera traps, vocalizations, and feces) have revealed a higher relative abundance index for Reeves’ muntjac (39.59%) than those for sika deer (3.90%) in the TNNR (Kong et al., 2024). The previous population of sika deer comprised only 365 individuals, with a growth rate of 17% in 1983, which is currently less than 2% (Jiang et al., 2012).

As ruminants, sika deer and Reeves’ muntjac have long food retention times in the digestive system, an aspect that imposes certain limitations on the use of traditional analysis. Direct tracking observation and indirect utilization were used in the diet study of Moschus chrysogaster (Zheng & Pi, 1979), Rucervus eldii hainanus (Song & Li, 1992), and grazing sheep (Lin et al., 2011). Ramirez, Quintanilla & Aranda (1997) estimated the selection of white-tailed deer (Odocoileus virginianus) based on microhistological analyzes of feces. Gebert & Verheyden-Tixier (2001) defined the food resources of red deer (Cervus elaphus) using stomach content analysis. However, the wary sika deer is difficult or impossible to observe directly. It is also difficult to obtain stomach samples from rare and endangered sika deer. Although faeces provide potential for ‘ecological detection’ on a tremendous variety of fronts but suffers from several problems due to differential digestibility, different size of particles and difficulties in identifying a large proportion of plant fragments in ruminant diet analysis (Putman, 1984).

Advances in sequencing technology have led to increased use of DNA-based diet determination, particularly DNA metabarcoding (Taberlet et al., 2012). DNA metabarcoding based on high-throughput sequencing allows simultaneous identification of mixed samples originating from multiple species (Li, Jiang & Chen, 2021). The method involves extraction of total DNA from fecal and stomach content samples, the polymerase chain reaction (PCR) amplification of DNA barcode markers from food taxa of interest, and then DNA sequencing for taxonomic classification of the recovered sequences (Deagle et al., 2019). Thus, food items can be accurately classified to the species level, enabling the identification of degraded or mixed dietary samples (feces, food boluses, or stomach contents) (Lenain, Olfermann & Warrington, 2004; Barco et al., 2016; Zhang et al., 2020b). Using high-throughput sequencing and DNA metabarcoding facilitates diet analysis, and the method can also compensate for the limitations of traditional methods in terms of qualitative and quantitative analyses (Pompanon et al., 2012).

Animal’s diet is an important attribute of its trophic niche and affects its role in the ecosystem. As such, diet can be used to gauge interspecies relationships (Du-Preez et al., 2017). Sika deer and Reeves’ muntjac are ruminants belonging to the Cervidae and may have similar diet selection requirements due to their evolutionary and physiological similarities (Schaller, 2000; Lv et al., 2020). Considered together with the fast reproductive cycle and dominant population of Reeves’ muntjac, this exerts interspecific pressure and potential resource competition (i.e., for space and food). However, several significant differences between the species may facilitate their coexistence, even if resources are limited (Glen & Dickman, 2008). Classical ecological theory offers two principal explanations for the coexistence of species in a community: habitat differentiation and resource differentiation (Shmida & Ellner, 1984). Species coexistence theory also emphasizes niche partitioning (Chesson, 2000), positing that interspecific competition typically arises when two or more species use the same resources, but the similarity of niches is limited (Chu et al., 2017). The strategic distribution of trophic resources plays a pivotal role in the mechanisms enabling the coexistence of sympatric herbivores with similar resource requirements (Filella et al., 2024). Therefore, we hypothesize that (i) Sika deer and Reeves’ muntjac would expand their diet breadths in summer, assuming that summer is when resources are most adequate; (ii) diet partitioning will increase with opportunity in summer, and perhaps weaken the diet niche overlap between two herbivores. Our study aims to investigate the dietary composition and nutritional niche overlap between sika deer and Reeves’ muntjac using high-throughput sequencing and DNA metabarcoding techniques. This information is significant to population conservation and management of sika deer and biodiversity monitoring.

Materials and methods

Study area and sample collection

The TNNR is located on the southern bank of the middle and lower reaches of the Yangtze River, Pengze, Jiangxi Province. The total area of the TNNR is 12,500 hm2, and the reserve is divided into three zones such as core, experimental and buffer. Most of the sika deer lived in the core zone. An experimental zone serves human activities and regulated development. A buffer zone with an area of 8,000 hm2 has some allowable human activities, thereby mitigating the effect of the human activity zone on the core zone (Liu et al., 2008). The TNNR is in a subtropical monsoon climate zone with four distinct seasons. Most plants begin to germinate during the spring. The summer vegetation type features mixed evergreen-deciduous broad-leaved forest, coniferous forest, mixed coniferous-broadleaved forest, broad-leaved forest, and bamboo, with abundant and nutrient-rich forage plants. The plant phenology enters a period of color change and leaf shedding in autumn. Especially in winter, deciduous broad-leaved forests become dormant; perennial and annual herbs wither, and the plant community structure and forest phase are prone to change.

Based on previous camera trap surveys, our sampling sites were largely set in areas with frequent activity of sika deer, i.e., Nursery bases, XianLingAn, fir forests, WuGuiShi, NieJiashan, and the Bamboo Garden. Three to five transects (2 km surveyed per transect) were set up at each sampling site, and each transect was randomly positioned in the study area (Fig. 2). To minimize the probability of multiple samples from the same individuals, all collected samples were separated by at least 30 m. To distinguish between the fecal pellets of sika deer and Reeves’ muntjac, fecal pellet dimensions are usually the best guide (Chapman, 2004). The fecal pellet morphology of sika deer is similar to that of black peanuts, while for Reeves’ muntjac, the fecal pellets are cylindrical and spherical with a smaller size (Cao et al., 2024). For samples collected from mixed-species flocks (including samples between adults of one species and juveniles of another), we used the COI gene fragment to identify the species. The fresh fecal samples were collected using sterile tweezers and transferred into sterile hermetically sealed bags, which were then transported at 4 °C to the laboratory and stored at −80 °C. A total of 60 fecal samples from two species (30 each) were collected in the summers of 2022 and 2023.

Figure 2 Sampling sites at the Taohongling Sika Deer National Nature Reserve.

(MP: Nursery bases; XLA: XianLingAn; SS: Fir forests; WGS: WuGuiShi; NJS: NieJiashan; ZY: Bamboo garden). Built map with ArcGIS Pro V3.0.0: https://www.esri.com/zh-cn/arcgis/products/arcgis-pro/overview.

DNA extraction and trnL amplification

The host and fecal plant DNA were extracted with a QIAamp Power Fecal DNA Kit (Qiagen, Hilden, Germany) and plant genomic extraction kits (Omega Bio-Tek, Norcross, GA, USA) according to the manufacturer’s guidelines. For DNA extraction in each round, negative controls (i.e., extraction without feces) were included to monitor for possible contamination. The DNA optical density value was measured using an ultraviolet spectrophotometer, and the A260/A280 ratio of most DNA extracts was between 1.70 and 2.21, indicating highly purified DNA. COI primers F: 5′-TTGGTGCCTGAGCAGGCATAGT-3′ and R: 5′-GAGAACAAGTGTTGATATAGAAT-3′ were used for amplifying, and species identification of herbivores (Zhang et al., 2011). The chloroplast trnL (UAA) intron was amplified with primers c: 5′-CGAAATCGGTAGACGCTACG-3′ and h: 5′-CCATTGAGTCTCTGCACCTATC-3′ (Taberlet et al., 2007). PCR amplifications were performed in a total volume of 25 μL of PCR mixture containing 12.5 μL of PCR mix (Tiangen, Beijing, China), 1 μL of DNA, 1 μL of each primer, and 9.5 μL of H2O, with a PCR negative control. The reaction conditions were as follows: denaturation at 95 °C for 5 min followed by 35 cycles at 95 °C for 30 s, 56 °C for 30 s, and 72 °C for 45 s, with a final extension at 72 °C for 10 min at and storage at 4 °C for 10 h. A PCR blank was included as a negative control, and no contamination was detected. The PCR products were detected using agarose gel electrophoresis for subsequent high-throughput sequencing.

Bioinformatic and statistical analyses

The valid fecal amplicons were purified and pooled for sequencing by Shenzhen Microsun Technology Co., Ltd., Guangdong, China. Paired-end sequencing was performed using the Illumina HiSeq X Ten system (Illumina Inc., San Diego, CA, USA). The raw data were processed using Trimmomatic (v1.2.11) and Flash software (v0.33). The barcoding at the end and the primer sequence distinguished the samples to obtain an effective sequence and correct the sequence direction, resulting in optimized data. After quality inspection and control of the original data, demultiplexed sequences from each sample were quality filtered and trimmed, denoised, and merged, and any chimeric sequences were identified and removed using DADA2 plugin in QIIME2. Each generated unique sequence was referred to as an amplicon sequence variant (ASV) at the 100% threshold of similarity. Representative sequences of the ASVs were selected and compared with the Nucleotide Sequence Database (NT) using a 99% sequence similarity threshold to obtain species annotation information by using the QIIME2 software.

To test the first prediction, the read abundance data were converted to relative read abundance (RRA, i.e., proportional summaries of counts) of each food item (Deagle et al., 2019). We also analyzed the intra- and interspecific differences in diet composition. Alpha diversity refers to diversity on a local scale, describing the species diversity (richness) within a functional community (Shannon, 1948; Andermann et al., 2022). Indices of diversity, including Observed_species, Shannon’s information index, Faith’s phylogenetic diversity (Faith’s_pd), and Pielou’s index, were used in the QIIME2 plugin to calculate alpha diversity. Kruskal-Wallis and Wilcox tests implemented in the QIIME2 software were used after obtaining the overall alpha diversity indices for statistical analysis and visualization of significant differences between groups. Afterward, the differences in food composition structure between groups were analyzed using permutational multivariate analysis of variance (PERMANOVA) implemented in the adonis function under the R package vegan version 4.3.3 and the “qiime diversity beta-group-significance” command in QIIME2.

To further validate our second prediction, dietary breadth was measured using Levins’ index (Levins, 1970), and the dietary overlap of each species was calculated using Pianka’s index (Smith, 1982; Pianka, 1973). Pianka’s niche overlap index >0.3 was considered a meaningful niche overlap between species, and a significant niche overlap was considered at a value >0.6 (Sun et al., 2022). We performed a nonmetric multidimensional scaling (NMDS) analysis based on the Bray–Curtis dissimilarity, using Phyloseq package in the R software (version 4.3.2; R Core Team, 2023). Patterns of diet composition and dietary niche overlap of sika deer and Reeves’ muntjac were visualized in two-dimensional space using the NMDS plots. Linear discriminant (LDA) effect size (LefSe) analysis was performed to obtain a ranking of abundant modules in the diet plant species for sika deer and Reeves’ muntjac. A size-effect threshold of 4.0 on the logarithmic LDA score was used for discriminative functional biomarkers. A network analysis was performed using igraph package in the R software (version 4.3.2) to reflect the interactions of species enriched in each sample group.

Results

High-throughput sequencing of trnL metabarcoding

The gel electrophoresis analysis revealed that four samples with low concentrations and weak bands were unusable for further. Therefore, this study focused on analyzing a total of 56 samples from sika deer (Group 1 = 28 samples) and Reeves’ muntjac (Group 2 = 28 samples). The 56 samples produced 1,339,361 valid amplified sequences by high-throughput sequencing, with an average of 23,917 valid sequences per sample. The total number of valid bases was 192,872,294, with the shortest sequence being 120 bp, the longest average read being 338 bp, and the total average length being 144 bp. The ASVs common to two sample sets as well as those specific to each species were identified to reflect the compositional similarity and differences at the ASV level. A total of 458 ASVs were identified; the sika deer group had 88 unique ASVs accounting for approximately 19.21%, while the Reeves’ muntjac group had 52 unique ASVs, accounting for 11.35%. The species shared 318 ASVs, accounting for approximately 69.43% of the total.

Alpha diversity and inter-group differences

The Observed species and Chao1 indices reflected the richness of ASVs in the samples. The highest community richness values were 99.46 ± 9.19 for sika deer and 71.21 ± 6.54 for Reeves’ muntjac. The average Chao1 for the sika deer group was 121.59 ± 11.63, while for the Reeves’ muntjac group, the average was 87.09 ± 8.19. The Shannon and Simpson indices showed that higher community diversity was observed for sika deer than for Reeves’ muntjac (Shannon index: sika deer = 2.81 ± 0.22 and Reeves’ muntjac = 2.38 ± 0.20, on average). Faith’s_pd is a diversity index calculated based on a phylogenetic tree. The index uses representative sequences of ASVs within each sample to calculate the distances used in constructing the phylogenetic tree. The average Faith’s_pd for the sika deer group was 4.73 ± 0.27, while for the Reeves’ muntjac group, this was 3.74 ± 0.23. Pielou’s index reflects the species evenness; the averages were 0.51 ± 0.04 for sika deer and 0.44 ± 0.04 for Reeves’ muntjac (Tables S1 and S2). The species-based rarefaction curves reached plateaus as the sample sequencing reads increased (Fig. 3).

Figure 3 (A) Box-plot of the alpha diversity index using Kruskal-Wallis and Wilcox tests. In each panel, the abscissa is the group, and the ordinate is the value of the corresponding alpha diversity index. (B) Alpha rarefaction curves: Observed species index and Shannon index.

*Significant different between groups (p < 0.05).

Diet composition

Both “occurrence” (i.e., presence/absence of taxa) and “RRA” approaches are semi-quantitative surrogates for the true diet. The error associated with weighted occurrence data stems from overestimating the abundance of rare items (Deagle et al., 2019). We used RRA, which provides a more accurate view of species’ diet than the frequency of occurrence, to summarize the dietary data (Hou et al., 2021). Ultimately, a total of 160 food items were identified in the feces of sika deer, comprising 149 genera in 79 families. A total of 155 food items comprising 146 genera in 76 families were identified for Reeves’ muntjac, indicating diverse diets of these two herbivores. The top 10 most abundant unique forage plants detected in sika deer were Zygnema sp., Trapa natans, Acer amplum, Syzygium grijsii, Citrus reticulata, Campylopus sp., Oplismenus sp., Kadsura longipedunculata, Hypericum sp., and Hibiscus syriacus (Table 1). In contrast, the top 10 most abundant unique forage plants among the Reeves’ muntjac samples were Morus alba, Picrasma quassioides, Strobilanthes sp., Perilla frutescens, Ailanthus altissima, Juglans sp., Clerodendrum cyrtophyllum, Pinus thunbergii, Staurastrum sp., and Patrinia villosa (Table 2). For the common forage plants consumed by sika deer and Reeves’ muntjac, the top 10 species with the highest relative abundance at the species level were Smilax china, Rubus spp., Loropetalum chinense, Sassafras tzumu, Phyllostachys edulis, Cunninghamia lanceolata, Alangium chinense, Rumex acetosa, Rhododendron simsii, and Rhus chinensis (Table S3).

Table 1 Annotation information of specific diet ASVs including ASVs ID, abundance, order, family, genus, and species for Sika deer.

ASVs ID	Abundance	Order	Family	Genus	Species	
OTU240	3,003	–	–	–	Zygnema sp.	
OTU56	1,831	Myrtales	Lythraceae	Trapa	Trapa natans	
OTU71	1,345	Sapindales	Sapindaceae	Acer	Acer amplum	
OTU32	684	Myrtales	Myrtaceae	Syzygium	Syzygium grijsii	
OTU30	220	Sapindales	Rutaceae	Citrus	Citrus reticulata	
OTU246	212	Archidiales	Leucobryaceae	Campylopus	Campylopus sp.	
OTU54	123	Poales	Poaceae	Oplismenus	Oplismenus sp.	
OTU95	121	Oxalidales	–	–	–	
OTU256	77	Archidiales	Leucobryaceae	Campylopus	Campylopus sp.	
OTU124	68	Austrobaileyales	Schisandraceae	Kadsura	Kadsura longipedunculata	
OTU223	60	Malpighiales	Hypericaceae	Hypericum	Hypericum sp.	
OTU57	53	Malvales	Malvaceae	Hibiscus	Hibiscus syriacus	
OTU136	49	Poales	Cyperaceae	Carex	Carex gibba	
OTU150	46	Proteales	Sabiaceae	Meliosma	Meliosma cuneifolia	
OTU168	46	Saxifragales	Haloragaceae	Gonocarpus	Gonocarpus sp.	
OTU117	41	Boraginales	Boraginaceae	Lithospermum	Lithospermum erythrorhizon	
OTU35	37	Araucariales	Podocarpaceae	Podocarpus	Podocarpus neriifolius	
OTU158	27	Poales	Poaceae	Digitaria	Digitaria sp.	
OTU91	27	Rosales	Rosaceae	Sibbaldianthe	Sibbaldianthe sp.	
OTU194	26	Pottiales	Bruchiaceae	Trematodon	Trematodon longicollis	
OTU174	23	Poales	Poaceae	Eleusine	Eleusine indica	
OTU1	21	Fabales	Fabaceae	Hylodesmum	Hylodesmum podocarpum	
OTU20	17	Rosales	Rosaceae	Duchesnea	Duchesnea indica	
OTU161	15	Cornales	Cornaceae	Alangium	Alangium sp.	
OTU212	14	Myrtales	Lythraceae	Lagerstroemia	Lagerstroemia indica	
OTU140	14	Malvales	Malvaceae	Melochia	Melochia corchorifolia	
OTU92	12	Asterales	Asteraceae	Sonchus	Sonchus asper	
OTU10	8	Lamiales	Lamiaceae	Phlomoides	Phlomoides umbrosa	
OTU99	6	Oxalidales	Oxalidaceae	Oxalis	Oxalis sp.	
OTU28	5	Gentianales	Apocynaceae	Trachelospermum	Trachelospermum jasminoides	
OTU55	5	Fabales	Fabaceae	Lotus	Lotus sp.	
OTU276	5	Sapindales	Anacardiaceae	–	–	
OTU173	5	–	–	–	Unknown phycophyta	
OTU123	4	Sapindales	Sapindaceae	Koelreuteria	Koelreuteria paniculata	
OTU73	4	Asterales	Asteraceae	–	–	
OTU258	4	Malpighiales	Euphorbiaceae	Mallotus	Mallotus sp.	
OTU5	3	Cornales	Cornaceae	Cornus	Cornus macrophylla	
OTU107	3	Eubryales	Bryaceae	–	–	
OTU217	2	Fagales	Fagaceae	Quercus	Quercus variabilis	
OTU113	2	Malvales	Bixaceae	Bixa	Bixa sp.	

Table 2 Annotation information of specific forage plants ASVs including ASVs ID, abundance, order, family, genus, and species for Reeves’ muntjac.

ASVs ID	Abundance	Order	Family	Genus	Species	
OTU377	1,503	Rosales	Moraceae	Morus	Morus alba	
OTU288	333	Sapindales	Simaroubaceae	Picrasma	Picrasma quassioides	
OTU314	255	Lamiales	Acanthaceae	Strobilanthes	Strobilanthes sp.	
OTU326	207	Lamiales	Lamiaceae	Perilla	Perilla frutescens	
OTU287	144	Sapindales	Simaroubaceae	Ailanthus	Ailanthus altissima	
OTU296	133	Fagales	Juglandaceae	Juglans	Juglans sp.	
OTU311	95	Lamiales	Lamiaceae	Clerodendrum	Clerodendrum cyrtophyllum	
OTU492	74	Pinales	Pinaceae	Pinus	Pinus thunbergii	
OTU497	62	–	–	–	Staurastrum sp.	
OTU337	21	Dipsacales	Caprifoliaceae	Patrinia	Patrinia villosa	
OTU520	19	Fabales	Fabaceae	Amphicarpaea	Amphicarpaea edgeworthii	
OTU346	11	Euphorbiales	Euphorbiaceae	–	–	
OTU350	9	Urticales	Moraceae	–	–	
OTU371	8	Ranunculales	Ranunculaceae	Clematis	Clematis florida	
OTU419	6	Ranunculales	Lardizabalaceae	Sargentodoxa	Sargentodoxa cuneata	
OTU489	6	Lamiales	Scrophulariaceae	Buddleja	Buddleja lindleyana	
OTU380	5	Ranunculales	Ranunculaceae	Ranunculus	Ranunculus japonicus	
OTU463	5	Caryophyllales	Caryophyllaceae	Pseudostellaria	Pseudostellaria heterophylla	
OTU500	4	Gentianales	Rubiaceae	Damnacanthus	Damnacanthus indicus	
OTU498	4	–	–	–	Unknown phycophyta	
OTU323	3	Lamiales	Oleaceae	Osmanthus	Osmanthus fragrans	
OTU376	3	–	–	–	Unknown bryophytes	
OTU324	2	Asparagales	Amaryllidaceae	Allium	Allium sativum	
OTU315	2	Ranunculales	Papaveraceae	Corydalis	Corydalis balansae	
OTU375	2	Cucurbitales	Cucurbitaceae	–	–	

Dominant genera and species in the complete diet spectra

Due to point mutations, multiple ASV representative sequences may belong to the same species, and these need to be merged into unique sequences. At the genus level, the dominant genera in both the sika deer and Reeves’ muntjac groups were Smilax (15.19%), Rubus (10.89%), Dicranum (8.74%), Loropetalum (5.97%), and Sassafras (4.74%) (Fig. 4). At the species level, the most dominant food item in the feces of sika deer is Smilax china (RRA = 24.45%), followed by Rubus spp. (~7.24%), Loropetalum chinense (~5.72%), Pohlia elongata (~5.07%), Cunninghamia lanceolata (~4.29%), and Rhododendron simsii (~3.68%). Rubus spp. (~14.75%), Dicranum scoparium (~14.45%), Sassafras tzumu (~9.44%), Loropetalum chinense (~6.50%), and Phyllostachys edulis (~5.12%) were the dominant food items for Reeves’ muntjac (Table 3). The distribution histograms of the top 20 species in the sika deer and Reeves’ muntjac groups are shown in Fig. 5. LEfSe analysis revealed the significant differences in forage plants between sika deer and Reeves’ muntjac (LDA score > 2.0, p < 0.05). Among those, three orders (Bryales, Asterales, and Liliales) and three families (Bryaceae, Asteraceae, and Smilaceae) were enriched in sika deer. Four orders (Cornales, Lamiales, Laurales, and Saxifragales) and five families (Dicranaceae, Cornaceae, Lauraceae, Moraceae, and Hamamelidaceae) occurred in Reeves’ muntjac (Fig. 6).

Figure 4 The bar chart distribution of the dominant forage plant in sika deer and Reeves’ muntjac groups at the genus level.

The x-axis stands for individual samples.

Table 3 Relative read abundance (RRA; %) of food item in the diets of Sika deer and Reeves’ muntjac (Top 20).

Number	Food items	Sika deer	Food items	Reeves’ muntjac	
1	Smilax china	24.45%	Rubus spp.	14.75%	
2	Rubus spp.	7.24%	Dicranum scoparium	14.45%	
3	Loropetalum chinense	5.72%	Sassafras tzumu	9.44%	
4	Pohlia elongata	5.07%	Loropetalum chinense	6.50%	
5	Cunninghamia lanceolata	4.17%	Phyllostachys edulis	5.12%	
6	Rhododendron simsii	3.77%	Smilax china	4.50%	
7	Persicaria perfoliata	3.25%	Alangium chinense	4.44%	
8	Dicranum scoparium	3.02%	Rumex acetosa	4.31%	
9	Erigeron annuus	3.16%	Premna microphylla	4.08%	
10	Rhus chinensis	3.15%	Glyphomitrium sp.	2.61%	
11	Setaria viridis	2.82%	Wisteria sinensis	1.92%	
12	Digitaria sanguinalis	2.29%	Broussonetia papyrifera	1.79%	
13	Phyllostachys edulis	2.19%	Platycarya strobilacea	1.83%	
14	Lespedeza bicolor	2.40%	Schima superba	1.83%	
15	Persicaria maculosa	2.15%	Ligustrum quihoui	1.55%	
16	Rosa laevigata	2.18%	Ligustrum quihoui	1.46%	
17	Bidens pilosa	2.16%	Cunninghamia lanceolata	1.45%	
18	Rubus coreanus	1.84%	Prunus mume	1.38%	
19	Carpesium abrotanoides	1.85%	Rubus coreanus	1.15%	
20	Oxalis corniculata	1.72%	Lophatherum gracile	1.00%	

Figure 5 Top 20 forage plants with the highest proportions in sika deer and Reeves’ muntjac groups at the species level.

The x-axis stands for individual samples.

Figure 6 (A) Cladogram based on LEfSe analysis, showing ASVs with the significance of two herbivores (green: sika deer; dark green: Reeves’ muntjac). (B) Log10-transformed LDA scores for ASVs, i.e., with a threshold value >4.0.

Interspecific niche partitioning and network analysis

Based on the NMDS analysis at the ASV level, the stress value of 0.208 indicated a good fitness of the NMDS model. There was a certain degree of partitioning in dietary habits between sika deer and Reeves’ muntjac. Each point in the plot represents a sample, and points shown in different colors belong to different sample sets. The distance between points represents the degree of community difference between samples. The closer the distance between two points, the higher the similarity in community structure and the smaller the difference. PERMANOVA detected significant differences between the food composition of sika deer and Reeves’ muntjac (PERMANOVA, p = 0.001, Pseudo-F = 5.17, R2 = 0.08, df = 1), supporting the results of the NMDS analysis. The niche breadth of a species reflects its degree of specialization. The wider the niche, the less specialized the species, indicating a tendency toward being a generalist. Conversely, a narrower niche indicates a tendency toward being a specialist. The niche breadths of sika deer and Reeves’ muntjac were 11.36 and 14.06, respectively. The dietary overlap index of the niches between sika deer and Reeves’ muntjac was 0.44, indicating that they share some food resources and have a moderate overlap in summer (Fig. 7A). The nutrients and plant secondary metabolites in forage plants (i.e., species and abundance) likely act in concert to alter the feeding habits of herbivores and foraging strategies (Villalba, Provenza & Bryant, 2002). Network tests showed the forage plant abundance between sika deer and Reeves’ muntjac at the genus level. Among these, Smilax was significantly correlated with Persea, Cinnamomum, and Alangium (p < 0.05). Rubus was significantly negatively correlated with Rhus (Fig. 7B).

Figure 7 (A) NMDS analysis of sika deer and Reeves’ muntjac with 95% confidence ellipse. (B) Network analysis of forage plants.

The size of the circles represents relative abundance, the lines indicate a significant correlation between two species (p < 0.05). Red lines mean positive correlations and blue means negative correlations.

Discussion

Multiple ASV representative sequences can be assigned to the same species based on the NT database, suggesting that there may be point mutations or next-generation sequencing errors among individuals within the plant species. Therefore, quantitative analysis of forage plants should be performed by merging and accumulation. For DNA identification of plants, researchers have proposed several combinations of DNA regions, i.e., rbcL + trnH-psbA, rbcL + ITS2, rpoC1 + matK + trnH-psbA, and rpoC1 + matK + rpoB (Pennisi, 2007). Additionally, the two-locus combination of rbcL + matK represents a pragmatic solution to a complex trade-off between sequence quality, discrimination, universality, and cost (Hollingsworth et al., 2009). However, despite a high separation rate obtained compared with a single gene barcode, only a plateau in resolution of ~70% was achieved from the plant dataset in combination (Fazekas et al., 2009). The rate of successful identification with ITS2 was 92.7% for medicinal plants, but the resolution of closely related species is still limited, especially within the species level (Chen et al., 2010).

The chloroplast trnL (UAA) gene selected was highly conserved in this study, and the amplification system and primers were robust and relatively specific, indicating a relatively good quantitative assessment of diet within and between species (Mallott, Garber & Malhi, 2018). However, some results obtained by alignment based on public databases are controversial. Interspecific hybridization and gene flow are quite common in plants, and some sequences may be difficult to identify to genus or species levels. Integrating the compound barcoding of trnL can improve the accuracy of species identification, i.e., the combination of trnL-trnF + ITS barcodes (Liu et al., 2018). In the field of dietary research, no universal primers are suitable for all taxonomic ranks due to varying recognition capacity, universality of DNA barcoding, and sequence variation across different plant taxa. Simultaneously, a local reference database of potential forage plants should be considered and constructed to provide sequence alignment resources and improve species identification derived from the reserve.

A previous study found that the diet of the South China sika deer comprised 37 plant species, containing 21 herbaceous and 16 woody species such as Smilax china, Rubus chingii, Rhododendron simsii, Rhus chinensis, and Cunninghamia lanceolata (Jiang, 2009). Smilax and Rubus were the dominant genera foraged by the two herbivores in this study. Smilax china is rich in nutrients, containing amino acids, fats, and organic acids, while extracts or active substances from Rubus spp. are also reported to have various pharmacological properties. Both of these plants are widely used in traditional Chinese medicine (Wang et al., 2023; Sheng et al., 2020). The functions of the nutritional and pharmacological components consumed from forage and their effects on the ruminants’ physiology need to be further explored. Additionally, more bryophytes were consumed by the two cervid species in summer, a finding that may be attributed to their preference for moist and shady valley habitats. In-depth monitoring is needed to confirm this intriguing phenomenon. To sum up, both South China sika deer and Reeves’ muntjac showed a preference for lianas and herbaceous plants. It has been speculated that different utilization patterns and co-evolution of food resources occur during long-term animal-plant interactions but not to the exclusion of the vegetation differences resulting from the subtropical and temperate marine climate (i.e., Japanese sika deer).

We detected interspecific differences in diet composition for sika deer and Reeves’ muntjac. The perennial vine Smilax china dominated in sika deer diet (24.45% RRA) but was just 4.50% RRA for Reeves’ muntjac. Rubus spp. and Dicranum scoparium together comprised 29.20% RRA for Reeves’ muntjac but just 10.26% for sika deer; Sassafras tzumu comprised 9.44% RRA for Reeves’ muntjac but just 0.11% for sika deer, indicating a certain level of dietary partitioning in their resources utilization. The differences in dietary species composition as plausible effect on coexistence because the interspecific competition may be relaxed relative to the scenario in which all herbivore species eat the same plant taxa, including the growth types (arbor, shrub and herbaceous) and family or genus (Pansu et al., 2022). For example, bison (Bison bonasus) consumed significantly more graminoids (21%), whereas legumes were more present in the sympatric fallow deer (Dama dama) diet (32%), this certain distribution of trophic resources between two species may facilitate their coexistence (Filella et al., 2024). Similarly, the selection of different food types by alpine musk deer (Moschus chrysogaste), red serow (Capricornis rubidus), and white-lipped deer (Przewalskium albirostris) helps avoid conflicts resulting from resource competition (Luo et al., 2024). In total, we found that both sika deer and Reeve’s muntjac selected a wide variety of plants in their diet. Although consumed common food items, differences in proportions occurred between the two species; furthermore, each species had exclusive plant species in summer, and the specific ASVs in sika deer were distinguished from those of Reeves’ muntjac.

Dietary selection and foraging strategies are affected by seasonal shifts, as animals consume different plants due to temporal and spatial changes (i.e., forest types, aspects, and physiognomy) in different seasons. Our prior winter data showed that sika deer predominantly foraged on Rubus spp., L. chinense, and Eurya japonica, accounting for 75.30% with a niche breadth of 4.53. Reeves’ muntjac mainly consumed Rubus spp., E. japonica, and Euonymus grandiflorus, representing 68.80% with a niche breadth of 3.44 (Wang et al., 2023). It is noteworthy that both sika deer and Reeves’ muntjac exhibited relatively broader dietary niches in summer (B = 11.36 and 14.06, respectively) and were generalist compared to winter. Previous studies showed that when forbs and new grasses were available to foraging deer, they would be expected to broaden their dietary niches to include forbs and thereby improve the diet quality (Nicholson, Bowyer & Kie, 2006). Our observations support this hypothesis that species expand their diet breadths in summer, and also indicate that diet selection and foraging strategies vary with food resource abundances and seasonal shifts (Nisha & Nishith, 2019).

Inter-specific competition may restrict the growth of the population, and sympatric species can achieve coexistence through niche separation to relieve substantial or potential competition (Lear et al., 2021). In the TNNR, two herbivores coexist sympatrically and share environmental resources, along with a short reproductive cycle and dominant population of Reeves’ muntjac, which may promote potential resource competition for sika deer (Jiang, 2009; Wang et al., 2023). However, a moderate degree of overlap was observed based on the results of nutritional niche overlap index (the Pianka index was 0.44). The dietary niche overlap may be affected by their diet partitioning in resources utilization (i.e., difference of food items and proportion). Similarly, sympatric roe deer (Capreolus capreolus), red deer (Cervus elaphus) and moose (Alces alces) with a moderate dietary niche overlap (52.6%) also showed differences in their proportion of each food type (Czernik et al., 2013). Furthermore, the nutritional ecological niche separation contributed to the stable coexistence among Wild yak (Bos grunniens), Tibetan wild ass (Equus kiang) and Tibetan antelope (Pantholops hodgsoni) in Tibetan Plateau (Shi et al., 2016), which indicated our second hypothesis.

Competitive interactions among herbivores are predicted to be severe between species that have the same feeding style and similar body weights; nevertheless, species may partition resources by size and energy requirements when body weights are different (Prins & Olff, 1998; Ritchie & Olff, 1999). Reeves’ muntjac is slightly smaller than sika deer, and as the two species in the reserve for several decades, we speculate that body size is also one of the reasons for dietary partitioning. However, quantitative analysis of the dietary richness and quality between different body sizes has not yet been performed; thus, this surmise must be interpreted cautiously. What’s more, our study used RRA to quantify the diet of two species; however, this method is still controversial. One reason is that herbivores have relatively long gut transit times that can impede DNA fragment amplification (Sakaguchi, 2003). An additional complicating factor is that herbivore guts have different digestion abilities for different plants. Woody stems contain more indigestible material than leaves or buds, and the plants or plant tissues that are more thoroughly digested may result in more thoroughly degraded DNA and therefore be underrepresented in the resulting sequence counts (Shipley et al., 1999; Stapleton et al., 2022). The continual advancement of sequencing technology may further improve the ability of metabarcoding to accurately assess diet composition. More studies on captive herbivores fed a known diet may also better explain sources of bias in sequence counts and refine ways to alleviate these effects.

Conclusions

This study indicated that the dietary niche overlap was moderate for two sympatric herbivores. The diet partitioning between two species is promoted by the different of food items and proportion, which ultimately reduces the overlap of dietary niches and helps avoid conflicts resulting from summer resource utilization in TNNR. Subsequent efforts should establish a complete local barcoding database, enhance the investigation of available foraging plants (especially Rosaceae and Smilacaceae), assess the biomass of foraging plants, and strengthen dynamic monitoring of herbivores. Additionally, artificial cultivation of preferred forage, habitat improvement, and reserve boundary adjustments should be considered if necessary.

Supplemental Information

Supplemental Information 1 Alpha diversity indices including Observed ASVs, Chao1, Shannon, Faith pd, Simpson, and Pielou of sika deer sample group.

Supplemental Information 2 Alpha diversity indices including Observed ASVs, Chao1, Shannon, Faith pd, Simpson, and Pielou of Reeves’ muntjac sample group.

Supplemental Information 3 Common ASVs between sika deer and Reeves’ muntjac with top 100 ASVs in abundance.

We are grateful to Xiaohong Liu, Yongjiang Chen, and Yulu Chen of Taohongling Sika Deer National Nature Reserve for their help in sample collection, and Dr. Cheng Huang for the help with the ArcGIS.

Additional Information and Declarations

Competing Interests

Author Contributions

DNA Deposition

Data Availability

The authors declare that they have no competing interests.

Ruofei Li performed the experiments, analyzed the data, prepared figures and/or tables, authored or reviewed drafts of the article, and approved the final draft.

Dandan Wang performed the experiments, analyzed the data, prepared figures and/or tables, authored or reviewed drafts of the article, and approved the final draft.

Zhiming Cao performed the experiments, analyzed the data, prepared figures and/or tables, and approved the final draft.

Yuqin Liu performed the experiments, prepared figures and/or tables, and approved the final draft.

Wenguo Wu analyzed the data, prepared figures and/or tables, and approved the final draft.

Wuhua Liu analyzed the data, prepared figures and/or tables, and approved the final draft.

Jianwen Zhan analyzed the data, prepared figures and/or tables, and approved the final draft.

Yongtao Xu conceived and designed the experiments, prepared figures and/or tables, authored or reviewed drafts of the article, and approved the final draft.

The following information was supplied regarding the deposition of DNA sequences:

The raw sequence data are available at NCBI Short Read Archive: PRJNA1110641.

They are also available at Figshare: Xu, Yongtao (2024). DNA metabarcoding reveals summer diet diversity and niche partitioning by two sympatric herbivores. figshare. Dataset. https://doi.org/10.6084/m9.figshare.25026785.v5.

The following information was supplied regarding data availability:

The raw measurements are available in the Supplemental Files.

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
