# Peer review of "DNA metabarcoding reveals diet diversity and niche partitioning by two sympatric herbivores in summer"

_PeerJ, doi:10.7717/peerj.18665_

## Round 0.1 · original submission · Major Revisions

Dear Authors,

After a thorough review by two reviewers, it has been determined that major revisions are necessary for your manuscript. Specifically, you need to provide more detailed explanations of the materials and methods, redesign the hypothesis and objectives, and exercise caution with the conclusions drawn from your results. Please address the observations and suggestions made by both reviewers to improve the manuscript and enhance its chances of being considered for publication in PeerJ.

Best regards,

Armando Sunny

·

Basic reporting

Overall, I think this article shows promise. However, I highly encourage the authors to work with a colleague who is proficient in English and familiar with the subject matter or consult a professional editing service as the manuscript really suffers from grammatical errors and sentences that are difficult to understand.
A few examples but there are numerous throughout the manuscript
L20: "positive transformation of vegetation" - what do you mean by this?
L29: ..."produced species-specific consumption and utilization for foraging plants except the common." I'm not completely sure what the point of this sentence is
L44: what are "external creatures"?

Introduction (some specifics)
L87-88: I'm not sure I completely follow this sentence but this type of analyses has been done on other primers besides trnL which you discuss in your discussion.
L102: you introduce a theory with little or no context here. I think you should probably define it in a sentence or two here to help the reader follow.

Figures: I think a number of your figures would be easier to read with larger axis labels such as Figure 2b, 2c, and 2d
Figure 3: what are the x-axis groups? Are those individuals or pooled samples? It is unclear and not discussed in the text as far as I can see and certainly not in the figure legend.

Discussion:
I think this could be edited quite a bit. It is hard to understand how some of the background directly applies to your research. Furthermore, I think there are far better examples you can find to compare the literature to your findings. They may not be from the same region but may be asking similar questions like diet differences between two overlapping ungulate species. I think this whole section would benefit from reorganization that does a better job of comparing previous findings from the literature to yours and how your results add to the understanding of the diet breadth of these two species and niche partitioning between them. I also think you spend quite a bit of time discussing seasonal changes in diet but you only addressed summer diet. I found it confusing to spend so much time on seasonal changes when you did not look at this.
L 303-304: Please define the terms you are using. What is a staple food vs emergency food? Are these terms found in the literature?
L308-309: I'm not sure where this sentence fits into the point you are making?
L327: "in terms of herbivourous" huh?

Experimental design

In general, I believe this study falls within the aims and scope of the journal.

I think the authors would benefit from describing their question in more detail. They mention one main hypothesis but I think it would be easier to understand if they broke this up into a number of predictions. It would make the reader have a clearer understanding of why they are utilizing each of the tests that the authors ran. It would very much improve the readability of the study.

I also think you need to better describe your study design especially for readers not familiar with your study area. What is the difference between the core, experimental, and buffer zones? I am unfamiliar with the differences in where your sampling sites. For instance what is the difference between XianLingAn and WuGuiShi? I don't know what those are whereas I assume a fir forest sampling area is dominated by fir trees while the bamboo garden is dominated by bamboo? Or are this arbitrary names applied to regions? Additionally how were your transects established? How long were they? Were they random or based on specific features? You collected fecal samples from two species, how can you tell the difference between them? Do they look different?

Methods:
I think in your statistical analysis section you could help your reader by again specifying which prediction each of your analyses is addressing?

Some specifics:
L194: "observed-species" should this be 'observed species'?
L205: "faith_pd" should this be Faith's phylogenetic diversity index? - this happens multiple times in the paper.

Validity of the findings

Results:
L208-211: What citation are you basing your statement that RRA is an accurate view of the species diet over frequency? How are you accounting for differences in fragment size and digestibility of different items that may influence RRA?
L271: I am not sure what you are saying here at all.

I had a really hard time following along with many of the results. I think the authors would help the reader by creating some tables summarizing all the tests evaluating diversity within and between species.

I think the conclusion section was the strongest section of the paper.

Additional comments

I commend the authors on completing this work. I know it took a lot of time and effort. I think you have all the components of a strong paper however it really needs a lot of editing and structural changes to make all that effort apparent and easy to interpret and understand for the reader. I also think it would be great if you explained a bit why this deer is unique/ecologically important if you can? Again, not everyone reading this may be familiar with this ecosystem so helping us understand it a bit more will help a lot. You mention they are IUCN as LC but are protected locally? Why this discrepancy?

Also, you looked at different regions (you mention them) but you don't seem to analyze if there are any differences? Are the same individuals traveling between your sampling sites? Do you think diets and potentially the amount of dietary overlap differs by location? Presumably the available vegetation differs.

Reviewer 2 ·

Basic reporting

The authors provide an interesting and insightful analysis of the diet composition and overlap of two species of deer that co-occur in conservation areas in China. The analysis involved an adequate number of samples, modern DNA-based methods, and several complementary bioinformatic strategies to support their analysis. I found the conceptual framework of the paper could benefit from a clearer description of the conceptual background and hypothesis. The results and discussion could have also been streamlined to focus on the most important findings, taking care not to draw conclusions that extend beyond the study itself.

I also suggest revising the abstract to explain field-specific jargon, such as what an ‘ASV’ or ‘alpha diversity’ is and how the number of ASVs might pertain to the ecology of the study system. For example, how broad is the study area and is it reasonable to expect that 458 ASVs would be recorded if an ASV is treated as approximately equivalent to a plant species? Similar advice would apply to other sections of the manuscript, especially the methods and results.

Experimental design

The authors note that fine-grained comparisons of foods eaten by the two cervid species in this study area are not well known and therefore we have a gap in knowledge about how their diets overlap. The hypothesis they pose is that they ‘may exhibit dietary overlap and partitioning in their trophic niches to achieve coexistence.’ This hypothesis could be better defined because overlap and partitioning are not mutually exclusive (i.e., diets can overlap and have differences, to various degrees). Additionally, evidence of overlap or partitioning (a short-term pattern) would not be sufficient to conclude they achieve coexistence (a long-term mechanism). A clearer hypothesis might focus on the ways these populations overlap or partition their diets. For example, what kinds of resources do they share vs. what kinds of resources would only be likely to get eaten by one or the other population?

Validity of the findings

In the abstract, summarizing statements elsewhere in the paper, the authors state that “we conclude that trophic niche partitioning may alleviate the competition and promote coexistence…” Unfortunately, the study does not present the right kinds of data to support this claim. Clearly, there is some degree of niche partitioning between the two species. This partitioning could reflect mechanisms that alleviate interspecific competition, as the authors suggest. However, it could also indicate the presence of extreme interspecific competition if one species is excluding another from utilizing some subset of beneficial resources. Inferences about competition and coexistence require experimental evidence that resources are limiting, that one species harms the other, and that each species can increase when rare (i.e., ‘the invisibility criterion’). The study presents a number of exciting results pertaining to the diet composition and overlap of these two consumer species; I’d encourage the authors to focus on explaining why these results are valuable more directly.

Additional comments

Line 140: these trnL primers amplify a section of DNA that varies considerably in length across taxa (>100 bp difference); please report the expected size range rather than stating ‘approximately 145 bp.’

Lines 212-254: the full-page list of species names and relative abundances is very dense and difficult to read. Rather than interrupting the flow of the paper, I suggest summarizing the main results in the text and presenting more of these details in a table.

Line 263: I could not find a description of the PERMANOVA in the Methods section, so details of that analysis should be clarified, but often these methods require reporting of ‘pseudo-F’ values (rather than true F values) as well as R-squared values and degrees of freedom.

---

## Round 0.2 · Major Revisions

Thank you very much for your revisions to the manuscript based on reviewer comments. As you will see from the reviewer comments in this round there are still concerns and suggested made by especially reviewer 2. Please carefully address their concerns in a revision of your manuscript.

·

Basic reporting

Thank you very much for addressing my previous comments. I think your manuscript has improved greatly. I still think the paper needs some careful editing to help facilitate the points your are trying to address. I have tried to point them out and make some suggestions. Please do not feel like you need to copy my suggestions but they are there to help guide you with my thoughts about how to improve your sentences.

Experimental design

I have no further comments. I am not an expert in working with DNA metabarcoding so I cannot comment on extraction methods and bioinformatics.

Validity of the findings

Conclusions are much better stated and easier to understand.

Additional comments

Thank you for your hard work revising your manuscript. I have made extensive comments in the attached PDF.

Reviewer 2 ·

Basic reporting

The authors' aims and objectives are clear and easy to understand, as is the overall workflow. The difficulty that the authors continue to face is that ecological concepts are presented in ambiguous and potentially misleading ways, or without enough context to properly interpret.

For example, in the abstract:
"vegetation succession has significantly reduced understory forage resources" -- this may be a true statement, but it's impossible to understand what support there is for the statement (what, specifically, was measured, where the measurements were taken, and how strong the statistically "significant" effect might have been)?

"... exhibit nutritional partitioning in their diets and that resource competition was moderate ..." -- as I previously commented on this statement, I can find no evidence in the paper that competition was measured. Yes, the taxonomic component of diets were compared. But the resulting patterns of overlap could be consistent with high, medium, or low levels of competition -- we simply can't know based on the available data.

Methods
"...amplify an approximately 150 bp region..." in the original manuscript, this statement read "... amplify an approximately 145 bp region..." My review pointed out that this is a length-variable marker and so the wide range of lengths is far more informative than an approximate number. The authors report in the Results section that their data span the range of 120-338 bp, but this is not the same as an expected range (and it is far too broad to refer to as approximately 150 bp).

Results:
The PERMANOVA results need to be reported together with both degrees of freedom (numerator and denominator) as well as R-squared values. In their response letter, the authors contend that the software they are using does not provide R-squared values; this could reflect a flaw in the software or simply some user error, but it is not true of PERMANOVAs as a statistical method.

Experimental design

The authors present a revised set of hypotheses, but unfortunately these center on the misguided notion that a measure trophic niche partitioning can support inferences about competition and coexistence with no additional lines of evidence.

Unfortunately, the experimental design cannot adequately address the hypotheses because of the following issues:

Hypothesis 1: The authors may find a pattern of diet partitioning in summer. But what would it take to falsify this or undermine its presumed relevance for competition and coexistence? Diet partitioning exists on a continuum from subtle to extreme. Theoretically, niche partitioning is most relevant for long-term coexistence when resources are limited (e.g., winter). And, as noted on my prior review, diet partitioning is a pattern that can be consistent with strong interspecific competition -- one species may displace the other from a more optimal resource space.

Hypothesis 2: there is no evidence to support 'adequate food resources' are available in summer, nor that they are unavailable in other seasons. As stated, this prediction also seems to be in conflict with hypothesis 1 (but they are not presented as alternative hypotheses): hypothesis 1 posits that diet partitioning will increase with opportunity in summer but, assuming that summer is when resources are most adequate, hypothesis 2 posits that diet species would expand their diet breadths and perhaps enhance overlap.

Validity of the findings

Many aspects of the primary results -- focusing on DNA-based dietary inferences -- may be within the realm of what is commonly acceptable in the field. With persistent technical errors in the reporting, I find that hard to judge, but I can trust that the quality of data will become clearer with further revision. However, the main conclusions surrounding diet overlap as a measure of competition -- moderate competition at that -- are presented without any specific evidence to support the claim. These types of claims would need to be removed from the manuscript and the authors would need to focus their interpretations on the available evidence to present valid findings.

---

## Round 0.3 · Minor Revisions

Thank you very much for your revisions! We very much appreciate the effort you have put into the manuscript. You will see that we have gotten some more comments back from one of the reviewers and I would like to ask you to address these in another revision.

·

Basic reporting

I think it is very much improved since the previous submission. However, there are still a few sections that need some re-working to make them more clear. See below:
L48 – 50. This seems like a very awkward sentence to me. I think you are trying to make a point to back up your statement that some animals are behaving strangely but I think you can do much better in this opening paragraph. Are you trying to write a paper about odd behaviors in relationship to diet? I don’t think so, so perhaps it would be better to alter this paragraph to more closely set up the problem/question(s) you want to address.
L68 – 73: seems like a run-on sentence. I would split it into two.
Paragraph starting on 78: you go into a lot of detail about Reeves’ muntjac but do not provide the same for the sika deer. I would provide both so you can compare and contrast since that is one of the main components of your paper.
L111: niche? Ecological niche? What kind of niche?
L125: what is a diet species?
L212: was NMDS analysis performed using vegan package? Please make sure you state what packages you used.
Paragraph 232: Please provide SE or 95% CI’s
L374-389: I think this sentence needs to be restructured to make your point stronger. The next sentence also feels like it comes out of nowhere. I would try to connect the two sentences. The next sentence about mongoose also feel like they are not connected to your point. I recommend stating the point you are trying to make and then use the examples to back it up. It often feels like you are using the example to state the main point which is hard to understand because your study was about deer.
L390: I would start it with “Competitive interactions among herbivores are predicted…..
L409: is awkward and could use some modifications

Experimental design

It seems fine as far as the design that I can comment on.

Validity of the findings

Findings seems clear and reasonable based on the data.

---

## Round 0.4 · accepted · Accept

Thank you very much for further revising your manuscript based on reviewer comments. I am happy to recommend accepting your manuscript now. Congratulations and thank you for choosing PeerJ!